# Functional Diversity of Mammalian Small Heat Shock Proteins: A Review

**DOI:** 10.3390/cells12151947

**Published:** 2023-07-27

**Authors:** Chaoguang Gu, Xinyi Fan, Wei Yu

**Affiliations:** 1Institute of Biochemistry, College of Life Sciences and Medicine, Zhejiang Sci-Tech University, Xiasha High-Tech Zone No.2 Road, Hangzhou 310018, China; chaoguanggu9@gmail.com; 2Faculty of Arts and Science, University of Toronto, Toronto, ON M5S1A1, Canada; xin.fan@mail.utoronto.ca

**Keywords:** sHSPs, functional diversity, phosphorylation, disease, mammalian

## Abstract

The small heat shock proteins (sHSPs), whose molecular weight ranges from 12∼43 kDa, are members of the heat shock protein (HSP) family that are widely found in all organisms. As intracellular stress resistance molecules, sHSPs play an important role in maintaining the homeostasis of the intracellular environment under various stressful conditions. A total of 10 sHSPs have been identified in mammals, sharing conserved α-crystal domains combined with variable N-terminal and C-terminal regions. Unlike large-molecular-weight HSP, sHSPs prevent substrate protein aggregation through an ATP-independent mechanism. In addition to chaperone activity, sHSPs were also shown to suppress apoptosis, ferroptosis, and senescence, promote autophagy, regulate cytoskeletal dynamics, maintain membrane stability, control the direction of cellular differentiation, modulate angiogenesis, and spermatogenesis, as well as attenuate the inflammatory response and reduce oxidative damage. Phosphorylation is the most significant post-translational modification of sHSPs and is usually an indicator of their activation. Furthermore, abnormalities in sHSPs often lead to aggregation of substrate proteins and dysfunction of client proteins, resulting in disease. This paper reviews the various biological functions of sHSPs in mammals, emphasizing the roles of different sHSPs in specific cellular activities. In addition, we discuss the effect of phosphorylation on the function of sHSPs and the association between sHSPs and disease.

## 1. Introduction

Heat shock proteins (HSPs) are molecular chaperones that are widely present in bacteria, plants, and animals, where they are principally induced in response to physiological and environmental stressors [1]. HSPs play an important role in cellular functions such as dynamic protein balance, signal transduction, apoptosis, cell growth, and differentiation [2,3]. There are many kinds of HSPs, with molecular weights ranging from 10 to 100 kDa. According to protein molecular weight and homology, HSPs can be divided into small HSPs (sHSPs), HSP40, HSP60, HSP70, HSP90, and large HSPs [4]. At present, HSP70 has received the most attention among proteins in the HSP family, followed by HSP60 and HSP90, while there is relatively little information on sHSPs.

Members of the sHSPs family share an α-crystal domain (ACD) with a length of approximately 90 amino acids, which is flanked by a variable hydrophobic N-terminal region (NTR) and a flexible polar C-terminal region (CTR) (Figure 1A) [5,6]. The ACD is highly conserved and rich in β-strands, which fold to form an IgG-like β-sandwich structure (Figure 1A) [7,8]. In addition, sHSPs have the ability to assemble into homo- or heteromeric dimers or oligomers to different degrees, and the oligomers usually have more than 12 subunits (Figure 1B) [9,10,11,12]. The formation of the oligomers is suggested to proceed via dimers and the subsequent assembly of tetrameric or hexameric substructures prior to the final polymerization of the oligomers (Figure 1B), which are mainly mediated by the ACD, the CTR I-X-I/V motif, and the NTR, respectively [13]. Interestingly, smaller species, such as dimers, are often active forms of sHSPs [14,15], indicating that the oligomers may serve as pools of inactive sHSPs. In mammals, a total of ten sHSPs (HSPB1~10) have been identified, four of which (HSPB1/5/6/8) are ubiquitously expressed in the human body, while the remaining six show tissue-specific expression patterns (Figure 1C) [16]. For example, HSPB4 is expressed mainly in the lens, while HSPB9 is expressed only in the testes. The most important characteristics of HSPB1~10 are listed in Table 1. sHSPs are capable of exerting functions under physiological, pathological, or stressful conditions. In addition to preventing the aggregation of unfolded proteins, sHSPs are involved in critical cellular activities such as apoptosis, autophagy, cytoskeleton maintenance, and differentiation, and their activity is regulated by phosphorylation cascades (Figure 1D,E) [17,18,19]. Because of their versatility, abnormalities in sHSPs are closely associated with human diseases such as neuropathy, oculopathy, and myopathy (Figure 1F) [20]. Here, we will elaborate on the various biological functions of sHSPs in mammals, emphasizing the roles of different sHSPs in specific cellular activities. In addition, we will discuss the effect of phosphorylation on the function of sHSPs and the association between sHSPs and disease.

## 2. sHSPs Are Involved in Diverse Vital Cellular Activities

### 2.1. sHSPs Prevent Protein Aggregation

sHSPs are ATP-independent chaperones whose primary function is to prevent the aggregation of unfolded proteins but not to fold these substrate proteins [57]. Accordingly, sHSPs are defined as “holdases”. HSPB4 and HSPB5 are highly expressed in the lens and maintain its transparency by protecting the lens proteins from irreversible aggregation [58]. sHSPs have different chaperone activities and substrate spectra, whereby HSPB1, HSPB4, and HSPB5 show higher chaperone activity in high-temperature-induced aggregation than other sHSPs [32]. By contrast, HSPB7 does not seem to inhibit the aggregation of unfolded proteins, but it is the most potent polyQ aggregation suppressor within the sHSPs family [45]. A recent study showed that HSPB1 interacts preferentially with polyQ-expanded mutant huntingtin protein (HTT) compared to normal HTT and affects its aggregation. Additionally, HSPB1 forms a complex with the autophagy cargo receptor protein sequestosome-1 (SQSTM1/p62) and regulates the unconventional secretion of HTT [59]. Under stressful conditions, sHSPs bind to the early unfolding intermediates of easily aggregable proteins to form stable sHSP-substrate complexes and maintain the folding capacity of the substrate [60,61,62]. Smaller species, including dimers, commonly prevent aggregation as active forms of sHSPs, possibly because smaller species have more unfolded structures in their subunits, which help enhance the chaperone activity of sHSPs [14,15,63]. Notably, some specific sHSPs are capable of forming heterooligomers, whereby the types and proportions of sHSPs in the heterooligomers affect their chaperone function [10,64]. Some studies have shown that activated HSPB1 and unfolded proteins form co-aggregates, which are smaller and more regular than those formed in the absence of HSPB1 [21]. These co-aggregates help HSP70 and its co-chaperones, HSP40 and HSP110, to extract and refold protein substrates, as well as release and recycle dimers of HSPB1 from the aggregates [21]. Taken together, these reports may help us elucidate a general strategy through which sHSPs prevent the aggregation of substrate proteins (Figure 2). Furthermore, only unfolded proteins are allowed to pass through the narrow proteasome, while aggregation-prone proteins are poor proteasomal substrates [65]. HSPB1 selectively increases the solubility of aggregation-prone HSPB5 mutants, which is conducive to their degradation through the ubiquitin–proteasome pathway [66].

### 2.2. sHSPs Suppress Apoptosis, Ferroptosis and Cellular Senescence

Apoptosis, a programmed cell death mechanism regulated by caspases, is utilized to remove unnecessary or damaged cells from the body and is mainly induced by extrinsic or intrinsic apoptotic pathways [67]. Apoptotic signal transduction pathways converge at the mitochondria, where they induce mitochondrial membrane permeabilization and the consequent release of cytochrome c, which is regulated by members of the B-cell lymphoma-2 (Bcl-2) family [68]. Among these proteins, some are antiapoptotic, like Bcl-2, Bad, and Bcl-XL, while others are pro-apoptotic, such as Bid, Bim, Bcl-XS, Bax, and Bak [69]. The balance between pro- and anti-apoptotic proteins determines cell death and survival [69]. Upon permeabilization of the mitochondrial membrane, cytochrome c is released into the cytoplasm and binds to apoptotic protease activating factor-1 (Apaf-1) to form the apoptosome. The latter then recruits and activates caspase-9, which in turn further cleaves and activates caspases 3, 6, and 7, leading to apoptosis [70]. sHSPs are widely involved in the regulation of critical events in the process of apoptosis (Figure 3), thus inhibiting apoptosis under many stressful conditions such as endoplasmic reticulum (ER) stress and photodamage [71,72].

In the extrinsic apoptotic pathways, tumor necrosis factor (TNF) death receptor superfamily member Fas activates apoptosis signal-regulated kinase 1 (ASK1) by recruiting its adapter DAXX, ultimately activating Jun N-terminal kinase (JNK) to induce apoptosis [73,74,75]. However, phosphorylated HSPB1 can inhibit the activity of DAXX and ASK1 by direct interaction [76,77], and HSPB6 was found to decrease ASK1 expression to inhibit apoptosis in epileptic rats [41]. Moreover, Fas is able to recruit and activate caspase-8 through another adapter, FADD, after binding to its ligand [78]. On the one hand, activated caspase-8 directly cleaves and activates caspase-3. On the other hand, caspase-8 cleaves and activates Bid, inducing apoptosis through the mitochondrial pathway [79]. Some studies have demonstrated that HSPB2 suppresses apoptosis by inhibiting the activation of caspase-8 [33].

In the intrinsic apoptotic pathways, in addition to directly binding and sequestering cytochrome c released into the cytoplasm, HSPB1 also suppresses apoptosis by regulating additional proteins, both upstream and downstream of cytochrome c [80]. A study found that HSPB1 is involved in MEK/ERK-mediated phosphorylation and proteasomal degradation of Bim to alleviate ER stress-induced apoptosis [71]. Another study showed that decreased expression of HSPB1 accelerates the intracellular redistribution of Bid from the cytoplasm to mitochondria [81]. As a downstream effector of Bim and Bid, activated Bax can oligomerize and translocate to the mitochondrial outer membrane to increase its permeability [82]. HSPB1 antagonizes Bax translocation by promoting the serine/threonine kinase AKT activation via a phosphoinositide 3 kinase (PI3K)-dependent mechanism [22]. HSPB1 also performs an anti-apoptotic function by preventing caspase-9/3 cleavage and activation after cytochrome c release [83,84]. Recent studies reported that the expression level of Bcl-2 is upregulated, whereas Bax and caspase-3 are downregulated following HSPB1 overexpression [85,86,87]. Moreover, the antiapoptotic effect of HSPB1 is associated with the p53/p21 pathway. Studies have shown that HSPB1 inhibition contributes to the nuclear accumulation of p21 through suppressing the phosphorylation of AKT, which results in apoptosis [72]. Similar to HSPB1, both HSPB4 and HSPB5 inhibit Bax, Bcl-XS, and caspase-3 [88]. In human and rabbit lens epithelial cells, HSPB5 abrogates ultraviolet-A (UVA)- or calcium-induced apoptosis by suppressing the RAF-MEK-ERK pathway and the expression of p53 and Bax [37,89]. Indeed, HSPB4 and HSPB5 have been reported to inhibit apoptosis through the AKT pathway in lens epithelial cells and astrocytes, respectively [37,90]. In addition to inhibiting apoptosis by activating the PI3K/Akt cell survival pathway, HSPB4 has also been shown to inhibit apoptosis by enhancing phosphorylation of Bad in lens fiber cells, and HSPB4’s antiapoptotic function is directly related to its chaperone activity [35,91]. Notably, both HSPB1 and HSPB5 are capable of reducing the generation of apoptosis-stimulating reactive oxygen species (ROS) to exert their protective effects [90,92,93]. Moreover, studies have demonstrated that HSPB6 regulates apoptosis by forming a complex with Bax [94]. HSPB8 is also one of the sHSPs with antiapoptotic activity that can counteract the effects of oxidative stress by regulating the mitochondrial pathway [49]. Recently, HSPB8 was described as an inhibitor of apoptosis in various injuries, such as hippocampal, myocardial, and early brain injuries [53,95,96].

Ferroptosis is a recently discovered programmed cell death mechanism driven by iron-dependent phospholipid peroxidation, and the past decade has witnessed a rapid development of research related to ferroptosis since it was first proposed in 2012 [97]. In the canonical ferroptosis pathway, the accumulation of phospholipid hydroperoxides (PLOOHs) leads to rapid and irreparable damage to the plasma membrane that can be ameliorated by the PLOOH-neutralizing enzyme glutathione peroxidase 4 (GPX4) [98]. HSPB1 facilitates the expression of glucose-6-phosphate dehydrogenase (G6PD) to augment GPX4 levels and lessen ferritin levels, thereby attenuating ferroptosis [23]. Furthermore, there is evidence that the Ser15 residue of HSPB1 is an important phosphorylation site in the protective response to ferroptotic stress [99]. Recently, HSPB1 has been identified as a regulatory hub of ferroptosis during the progression of hepatocellular carcinoma [100].

Cellular senescence was originally defined as the phenomenon of significant changes in cell morphology and metabolic activity due to limited cell replication capacity, which is attributed to the shortening of telomeres [101]. At present, senescence is also considered a stress response to internal or external stimuli, including oxidative stress, irradiation, and mitochondrial dysfunction [102]. Senescence-associated growth arrest depends on the activation of the p53/p21, and p16 signaling pathways [103]. Studies have shown that down-regulating HSPB1 increases the expression of p53 and p16, while expression of HSPB1 at elevated levels leads to reduced accumulation of p21 [24,104]. Interestingly, expression of human HSPB1 in yeast extends its replicative lifespan [105]. In addition, expression of the naked mole-rat HspB1 also improves lifespan and enhances stress resistance in *Caenorhabiditis elegans* model [106].

### 2.3. sHSPs Promote Autophagy

Autophagy is a pathway by which cells degrade misfolded proteins and damaged organelles to maintain intracellular homeostasis under stressful conditions such as starvation and oxidative stress [107,108]. Autophagy is traditionally classified into three main categories: macroautophagy, microautophagy, and chaperone-mediated autophagy, among which macroautophagy is the most extensively studied, including the formation of autophagosomes and their subsequent fusion with lysosomes [17]. Autophagy-related genes (Atg) are key executive factors of autophagy, and Beclin-1 was the first mammalian Atg to be discovered [109]. Autophagosome formation involves the microtubule (MT)-associated protein light chain 3 (LC3), a mammalian homologue of yeast Atg8, and the conversion of LC3-I into its lipidated form LC3-II is important for the development of autophagy [110]. As a consequence, LC3-II is widely used as a marker of autophagosomes. Autophagy receptor SQSTM1/p62 is able to bind to LC3 and recognize ubiquitinated proteins to serve as an adaptor in autophagosomes [111]. Degradation of SQSTM1/p62 can be mediated by autophagy, and inhibition of autophagy will lead to the accumulation of SQSTM1/p62 [112]. HSPB1/6/8 was reported to participate in the process of macroautophagy (hereafter called simply autophagy) and regulate autophagy through various signaling pathways (Figure 4).

HSPB1 facilitates the formation of SQSTM1/p62 bodies by binding to SQSTM1/p62, which is essential for the nucleation of phagophores that are capable of elongating to become mature autophagosomes [25]. During autophagy, autophagosomes transport from the cytoplasm to the perinuclear region along MTs, ultimately reaching perinuclear lysosomes for degradation [113]. HSPB1 was demonstrated to favor the formation of non-centrosomal MTs, but its binding to MTs appears to be weak and transient [114]. In contrast, the S135F missense mutant of HSPB1 has a higher affinity for SQSTM1/p62 and α-tubulin, which blocks the MT-dependent transportation of autophagosomes and has a negative effect on its function of promoting autophagy [25,113]. One study reported that sustained HSPB1 expression leads to ER stress, which subsequently activates the AMP-activated protein kinase 1 (AMPK1)/unc-51-like kinase 1 (ULK1) pathway [115]. In addition, its expression leads to the accumulation of AKT1/glycogen synthase kinase 3 beta (GSK3B)-mediated SH3-domain GRB2-like B1 (SH3GLB1), thus triggering astroglial autophagy [115]. Notably, the same report mentioned that HSPB1 expression induces ER stress and activates the eukaryotic translation initiation factor 2 subunit alpha (EIF2S1)/activating transcription factor 4 (ATF4) pathway. Controversially, HSPB1 was described as acting as a negative regulator of ER stress and the EIF2S1/ATF4 pathway in autophagy induced by foot-and-mouth disease virus infection and inhibition of the proteasome [116,117]. The activated EIF2S1/ATF4 pathway can inhibit the AKT/mTOR pathway, thereby promoting autophagy [116]. In addition, HSPB1 was proven to promote Beclin-1-dependent autophagy by facilitating JNK phosphorylation in atherosclerotic vascular smooth muscle cells [86]. Moreover, studies have shown that HSPB1 regulates autophagy in glioblastoma cells through Atg7 [118]. In renal tubular cells, HSPB1 also increases LC3-II protein levels and decreases SQSTM1/p62 protein levels [87]. HSPB8 participates in autophagy as a component of the chaperone-assisted selective autophagy (CASA) complex, and CASA is a form of macroautophagy that initiates the ubiquitin-dependent autophagic sorting of damaged proteins for lysosomal degradation [51,119,120]. The CASA complex consists of HSP70, HSPB8, and Bcl-2-associated athanogene 3 (BAG3), the latter of which is a co-chaperone that regulates the ATPase activity of HSP70 and also serves as a scaffold to link HSP70 to HSPB8 [121,122,123]. Notably, SQSTM1/p62 is capable of interacting with the polyubiquitin chain of the protein substrates, which is ubiquitinated by the E3 ubiquitin ligase STUB1 after binding to the CASA complex [123]. Some studies have demonstrated that HSPB8 is essential for the fusion of autophagosomes with lysosomes, exerts the effects of preventing oxidative stress, and responds to high glucose by regulating autophagy [50,124,125]. Other studies have reported that HSPB6 competes with Bcl-2 to combine with Beclin-1, which prevents Beclin-1 from being degraded by the ubiquitin–proteasome pathway, thereby stimulating autophagy [42].

### 2.4. sHSPs Modulate Cellular Cytoskeleton and Membrane

The cellular cytoskeleton is a complex fibrous grid structure involved in the maintenance of cell morphology, division, movement, intracellular material transport, and other important cellular life activities [126]. It is well established that the cytoskeleton mainly consists of three structural components, including microfilaments (MFs), MTs, and intermediate filaments. sHSPs are closely associated with the cytoskeleton and respond to external stimuli by regulating its remodeling and reinforcement processes. Previous studies revealed that unphosphorylated monomers of HSPB1 are effective in inhibiting actin polymerization, whereas phosphorylated monomers or unphosphorylated oligomers are ineffective [127]. Inversely, phosphorylation of HSPB1 by the p38/MK2 pathway leads to the disassociation of monomeric actin (G-actin) from HSPB1 and subsequent polymerization [128]. Furthermore, phosphorylated HSPB1 is recruited to high-tension structures in cells under stretching stimulation, which is indispensable for actin cytoskeletal remodeling and reinforcement [129,130]. This protective effect of HSPB1 on the actin cytoskeleton appears to be indirect, as some evidence indicates that HSPB1 is a weak MFs side binding protein that predominantly interacts with partially denatured but not native actin to prevent aggregation of MFs [131,132]. Moreover, sHSPs can interact with various MFs-related proteins and influence their function, thereby playing a key role in the regulation of the contractile movement of muscle cells (Figure 5). It is well known that thick filaments made of myosin and thin filaments made of actin are arranged orderly in sarcomeres, which are the basic units that mediate the contraction of muscles. Titin is an essential component of the sarcomere and affects the elasticity of striated muscle cells [133,134]. Several studies revealed the transfer of HSPB1 and HSPB5 from the abundant cytoplasmic pool to titin’s unfolded Ig region under stretching stimulation, thereby preventing its aggregation and maintaining normal function [26,135]. However, the binding of sHSPs to titin contributes to an increase in the passive tension of muscle fibers, which is a symptom of myopathy [135]. Filamin c (FLNC) cross-links MFs into a network structure and is associated with myofibrillar Z-discs and intercalated discs in striated muscle [136,137]. HSPB1 and HSPB7 have been shown to directly interact with FLNC to modulate its localization, unfolding, and aggregation, which are closely related to the occurrence and progression of myopathy [46,137,138]. Notably, although the recognition of FLNC by HSPB1 is not phosphorylation-dependent, the stabilization of unfolded intermediates is regulated by the phosphorylation of HSPB1 [138]. Interestingly, the degradation of impaired FLNC as well as the subsequent stimulation of FLNC gene transcription by the transcriptional coactivators YAP and TAZ are mediated by the CASA complex and BAG3 [51]. HSPB7 has been identified as a negative regulator of the MF’s length that represses actin polymerization by directly binding to G-actin [139]. In addition, the absence of HSPB7 results in up-regulation of the actin nucleator leiomodin2 expression and mislocalization of the pointed end-capping protein tropomodulin 1, which result in an increase in F-actin length [139]. HSPB6 might indirectly affect the actin depolymerizing factor cofilin by interacting with the universal adapter protein 14-3-3 [43,140]. Furthermore, it was suggested that desmin, vimentin, glial fibrillary acidic protein, and other cytoskeletal proteins may also be protected by sHSPs [141].

There is increasing evidence that sHSPs are associated with the cellular biological membrane. HSPB1 and HSPB5 interact with lipid membranes and contribute to plasma membrane stability [27,38]. Interestingly, sHSPs have been shown to interact with many membrane proteins, which is critical to the function of the client proteins. For example, HSPB1 interacts with the autophagic protein Atg9a and regulates the induction of ER-specific autophagy [142]. Moreover, HSPB5 has the ability to rescue the transport of the mutated cystic fibrosis transmembrane conductance regulator to the membrane [143]. Notably, sHSPs are able to escape the cytosol and gain access to the extracellular environment, acting as signaling agents as well as trafficking between cells, which may be related to their interactions with membranes and membrane proteins [59,144].

### 2.5. sHSPs Regulate Cell Differentiation, Angiogenesis, and Spermatogenesis

The effects of sHSPs on cell differentiation are pleiotropic, and the underlying mechanisms remain elusive. HSPB1 appears to promote osteogenic differentiation of human adipose-derived stem cells (hASCs) and bone marrow-derived mesenchymal stromal cells (hBMSCs) through interactions with the cytoskeleton [28]. However, HSPB7 knockdown significantly promotes the osteogenic differentiation of hASCs while decreasing the osteogenic differentiation of hBMSCs [47,145]. Studies have shown that HSPB1 promotes epidermal differentiation and keratinization by regulating the processing of filaggrin [146,147]. HSPB1 was also shown to be essential for the differentiation of trophoblast cells and extravillous trophoblasts during placental development [148,149]. Moreover, knockdown of HSPB1 was found to promote the differentiation of dental pulp stem cells into oligodendrocytes [150]. HSPB3 was demonstrated to promote myogenesis by regulating the lamin B receptor [34]. The ACD of HSPB8 has been suggested to act as a factor of differentiation in adult hippocampal neurogenesis [52].

sHSPs are closely related to angiogenesis. Studies have shown that HSPB1 released from endothelial cells inhibits angiogenesis through direct interaction with vascular endothelial growth factor (VEGF) [29]. Interestingly, VEGF is able to reduce HSPB1 secretion through the p38/MK2 pathway-mediated phosphorylation of HSPB1 when angiogenesis is required [29,151,152]. Meanwhile, phosphorylated HSPB1 leads to actin polymerization and promotes reorganization of the actin cytoskeleton, which facilitates angiogenesis [151]. As a chaperone of VEGF-A, HSPB5 promotes its expression and secretion [39]. In addition, HSPB6 secreted through exosomes has been demonstrated to regulate myocardial angiogenesis through activation of the VEGF receptor 2 and improve cardiac function [44,153].

More studies are needed to clarify the roles of HSPB9 and HSPB10, which have previously been reported to be expressed in the testes [154]. However, a recent paper reported that HSPB10 is also expressed in the renal collecting ducts of rats [155]. HSPB9 and HSPB10 affect sperm production and male reproductive health [55]. Studies have shown that HSPB10 is essential for the connection between the head and tail of spermatozoids, which can be damaged by the disruption or haplo-deficiency of HSPB10 [56,156,157].

### 2.6. sHSPs Attenuate Inflammation and Oxidative Damage

sHSPs also show anti-inflammatory effects, reducing the level of pro-inflammatory cytokines during inflammation. In microglia, HSPB1 has been demonstrated to mediate the autophagic degradation of IκB kinase β through direct interactions, thereby reducing TNF-α expression to exert an anti-inflammatory effect [158]. It is suggested that HSPB1 also acts as an extracellular mediator to increase the expression of interleukin-10, an anti-inflammatory factor, by activating nuclear factor κB (NF-κB) in macrophages, though NF-kB is often regarded as having a pro-inflammatory role [30]. HSPB4 and HSPB6 suppress inflammatory responses by inhibiting NF-κB and activating the cAMP/PKA pathways, respectively [36,41]. Moreover, HSPB5 and HSPB8 have been reported to ameliorate inflammation of the kidneys and heart muscle [53,159].

At least some sHSPs, especially HSPB1/5/7/8, have been demonstrated to respond to oxidative stress and prevent oxidative damage [31,40,48,54]. This protective function is more common in the cardiovascular system, where mitochondrial activity is relatively high, as the mitochondria inevitably produce ROS that cause oxidative damage when generating energy [160]. Under oxidative stress, the oxidized Cys137 of HSPB1 forms disulfide bonds that lock the dimeric structure of HSPB1 to an increased, accessible hydrophobic surface, which facilitates the interaction between HSPB1 and client proteins [161]. Interestingly, the redox oscillation of HSPB1 exhibits a circadian rhythm, which is associated with higher rates of acute myocardial infarction in the morning [31]. However, sHSPs do not act as direct ROS scavengers but instead attenuate oxidative stress by increasing G6PD activity and the glutathione/oxidized glutathione ratio [162,163,164].

## 3. Phosphorylation Regulates the Biological Functions of sHSPs

Post-translational modifications are able to affect the function of proteins by altering their spatial structure and charge properties. sHSPs have been demonstrated to undergo various post-translational modifications such as phosphorylation [130], glycosylation [165], acetylation [166], deamidation and Asp racemization [167], as well as S-thiolation [168], among which phosphorylation is the most extensively studied. Half of the known sHSPs have been reported to be modified through phosphorylation, and phosphorylation of HSPB1 is the most extensively studied. Human HSPB1 has three phosphorylation sites: Ser15, Ser78, and Ser82 [169]. By contrast, there are only two modification sites in mice and rats: Ser15 and Ser86, where the latter is homologous to human Ser82 [170]. Phosphorylation of HSPB1 is regulated by numerous kinases and pathways, among which the p38/MK2 and PI3K/AKT pathways, as well as ERK1/2, have been widely reported in recent years [169]. In addition, HSPB1 is also phosphorylated by the cGMP-dependent protein kinase [171]. The Ser19, Ser45, and Ser59 residues of HSPB5 are phosphorylated via the p38/MK2 pathway [143,172,173]. Moreover, the cAMP/PKA and cGMP/PKG pathways are involved in the phosphorylation of Ser16 of HSPB6 [41,174]. In vitro, HSPB8 has been shown to be phosphorylated by PKC at residues Ser14 and Thr63, as well as by ERK1 at residues Ser24, Ser27, and Thr87, and to a lesser extent by casein kinase 2 [175,176]. cAMP-dependent protein kinase can also phosphorylate HSPB8 at residues Ser24 and Ser57 [177]. Furthermore, HSPB4 is phosphorylated at Thr148 [36].

The phosphorylated form is usually the activated form of sHSPs under stressful conditions and plays an essential role in the execution of their protective functions (Figure 6). When phosphorylated, sHSPs tend to change from oligomers to smaller species, which facilitates their chaperone function [178,179,180,181,182]. Some studies have shown that phosphorylated HSPB1 antagonizes apoptosis that is induced by TNF-α and TNF-related apoptosis-inducing ligands by promoting the activation of pro-survival signaling pathways, including the TAK1-p38/ERK and SRC-AKT/ERK axes [183,184]. Other experiments have also demonstrated the importance of sHSPs’ phosphorylation for their antiapoptotic function [41,85,185,186]. Blockade of HSPB6 phosphorylation at Ser16 was proven to suppress autophagy, thus exacerbating cardiac ischemia/reperfusion injury [187]. HSPB1 phosphorylation is imperative for its response to mechanical stress [129,130]. A study showed that phosphorylation results in increased exposure of residues surrounding the phosphorylated site of HSPB1, facilitating the interaction of HSPB1 with FLNC [138]. Under the stimulation of mechanical forces, HSPB1 is phosphorylated and translocated into the hypertonic structure of cells, participating in the remodeling and reinforcement of the cytoskeleton, thereby affecting cell diffusion and migration [19,129,130]. Complementing this, a recent study has shown increased levels of phosphorylated HSPB1 in the myometrium of rats during delivery [170]. Furthermore, phosphorylation has also been reported to promote the anti-ferroptotic and anti-inflammatory functions of sHSPs [41,99]. Interestingly, phosphorylation appears to influence the unconventional secretion of sHSPs, which is independent of the ER–Golgi pathway [188]. There is evidence that HSPB5 diverts autophagosomes toward the extracellular space rather than lysosomes [189]. However, phosphorylation of Ser59 inhibits the unconventional secretion of HSPB5 by inhibiting its recruitment to autophagosomes [189]. In addition, inhibition of HSPB1 phosphorylation weakens the osteogenic differentiation of hASCs and hBMSCs [28]. AKT and P38/MK2-mediated phosphorylation are critical for HSPB1-regulated differentiation of epidermal and trophoblast stem cells, respectively [146,147,149]. Moreover, studies have shown that the expression and phosphorylation levels of HSPB1 change with lens cell differentiation [190].

## 4. sHSPs and Disease

Because sHSPs perform a variety of important physiological functions, their abnormalities inevitably contribute to various diseases. Mutations in HSPB1, HSPB3, and HSPB8 cause motor neuropathies such as Charcot–Marie–Tooth disease and distal hereditary motor neuropathy, whereas mutations in HSPB5 are usually associated with myopathies [191,192,193,194,195]. Notably, more than 30 disease-related mutations in HSPB1 have been identified [196]. Furthermore, sHSPs are capable of preventing the accumulation of easily aggregated pathogenic proteins, such as Aβ, Tau, α-synuclein, and HTT, which can result in neurodegenerative disorders such as Alzheimer’s, Parkinson’s, and Huntington’s disease [197]. In addition to preventing the aggregation of pathogenic proteins, HSPB1 and HSPB8 are also suggested to ameliorate the severity of amyotrophic lateral sclerosis, a fatal neurodegenerative motor neuron disease, by regulating apoptosis and autophagy [198]. HSPB1, HSPB4, and HSPB5 are essential for maintaining the balance of proteins in the eye, and abnormalities in their function are associated with ocular diseases such as cataract [199,200]. Moreover, studies have shown that HSPB1 is involved in the development of cardiovascular diseases by affecting apoptosis, autophagy, oxidative stress, and inflammatory responses [17,201]. The mechanisms through which mutant sHSPs cause disease are complex. On the one hand, mutations may affect the structure and properties of sHSPs themselves, leading to the formation of larger oligomers of sHSPs and interfering with their phosphorylation [202]. On the other hand, mutations generally reduce the chaperone activity of sHSPs, resulting in the aggregation of substrate proteins [197]. Although some mutants, such as HSPB1 S135F, show higher binding affinity for client proteins, they are also impaired in their functions [25,113]. Interestingly, the activity of sHSPs is beneficial in the diseases mentioned above but counterproductive in cancer. sHSPs have been revealed to be overexpressed in multiple types of cancers, such as breast cancer [203], gastric cancer [204], cervical cancer [205], lung cancer [206], and glioblastoma [207]. Most of the sHSPs possess antiapoptotic activity and inhibit apoptosis in various malignancies [208]. However, a few studies have shown that certain sHSPs can suppress tumors. For example, the expression of HSPB6 has a pro-apoptotic effect in colorectal cancer [209]. In addition, increasing evidence has demonstrated that sHSPs promote the migration, invasion, and angiogenesis of cancer cells, as well as play a pivotal role in chemotherapy resistance [210,211,212].

It is worth mentioning that the protein transduction domain (PTD) appears to provide a therapeutic strategy for certain diseases associated with sHSPs mutations, in which the function of sHSPs is absent. sHSPs and peptides derived from sHSPs that are conjugated to the PTD of the HIV-1 trans-activator of transcription protein (TAT) are able to be delivered directly into cells and tissues [213]. TAT-HSPB1 is capable of suppressing ROS production and caspase activation in vitro as well as reducing myocardial infarction and improving cardiac function in vivo [214,215]. Similarly, TAT-HSPB8 was also proven to inhibit oxidative stress-induced apoptosis in hippocampal neuronal cells [49]. Furthermore, TAT-HSPB1_65–90_ peptide attenuates neurological deficits and cortical apoptosis after experimental subarachnoid hemorrhage (SAH) [216]. Surprisingly, a study showed that the TAT-HSPB1_12–35_ peptide induces lysosomal membrane permeabilization and the release of cathepsin D from lysosomes after entering clear cell renal cell carcinoma cells, resulting in apoptosis and inhibition of proliferation, which indicates that TAT-HSPB1_12–35_ is a potential therapeutic agent for renal cancer [217]. In addition, AZX100, a phosphorylated peptide analogue derived from HSPB6 that attaches to the PTD, has been demonstrated to reduce transforming growth factor-β1-induced connective tissue growth factor expression in keloid fibroblasts and prevent decreases in cerebral perfusion after experimental SAH [218,219].

## 5. Conclusions and Prospects

sHSPs are ubiquitous in all known organisms, sharing conserved ACD combined with variable CTR and NTR. sHSPs usually exist in the form of oligomers and depolymerize into smaller species to participate in various cellular activities under stressful conditions. A total of 10 sHSPs have been identified in mammals that bind to unfolding proteins in an ATP-independent mechanism, preventing their uncontrolled aggregation. In addition to chaperone activity, sHSPs also maintain cytoskeleton and membrane stability, inhibit apoptosis, ferroptosis, and senescence, promote autophagy, and alleviate inflammatory responses and oxidative damage, thereby promoting cell survival under stressful conditions. Moreover, sHSPs play a key role in cell differentiation, angiogenesis, and spermatogenesis. Phosphorylation usually produces an active form of sHSPs. The relationship between different functions of sHSPs is intimately interwoven, and several functions are generally induced in response to a common stimulus. For instance, damaged mitochondria are major sources of ROS that can cause oxidative damage [220,221]. In response to mitochondrial damage, sHSPs-mediated oxidative stress, mitophagy, and anti-apoptosis may function simultaneously [90,163,220]. The correlation between sHSPs’ functions is partly attributed to the fact that they share certain upstream or downstream signaling pathways. As mentioned above, HSPB1 phosphorylated by the p38/MK2 pathway favors actin polymerization and angiogenesis, while the inhibition of the p53/p21 pathway by HSPB1 is conducive to the suppression of apoptosis and senescence. Because sHSPs have diversified essential functions, when mutated, they often cause a variety of diseases in humans.

Despite significant achievements, there are still many areas regarding sHSPs that require further study. Recently, increasing studies have reported the extracellular presence of sHSPs (including HSPB1, HSPB5, and HSPB6), which are suggested to be released into the extracellular environment by lysosomes and/or exosomes [222]. Extracellular HSPB1 was shown to act as a signaling molecule and regulate the intracellular NF-κB pathway through toll-like receptors [223,224,225]. However, more research is needed to elucidate the mechanisms of sHSP secretion and identify their cell surface receptors and intracellular signaling pathways. In addition, there is limited information on the relationship between sHSPs and viral infections. Thus, it remains an open question whether sHSPs are normally hijacked by viruses and utilized for viral proliferation, similar to other HSPs [226]. Currently, research on the relationship between sHSPs and disease is focused on explaining how sHSPs contribute to the occurrence and development of disease, whereas there is a lack of sHSP-based therapy in various disease models. sHSPs show great therapeutic potential, so drug design targeting sHSPs and sHSP-associated signaling pathways deserves more attention.

## Figures and Tables

**Figure 1 cells-12-01947-f001:**
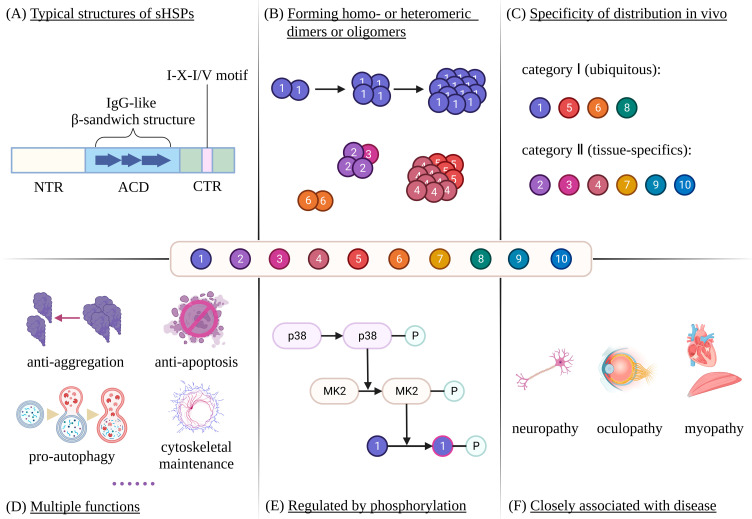
Characteristics of sHSPs in mammals: The figure illustrates several characteristics of sHSPs in mammals. Firstly, sHSPs share a conserved ACD, which are flanked by variable NTR and CTR (**A**). In addition, sHSPs are capable of forming homo- or heteromeric dimers or oligomers, whereby the latter are formed through multiple steps (**B**). HSPB1/5/6/8 are ubiquitously expressed in the human body, whereas HSPB2/3/4/7/9/10 show tissue-specific expression patterns (**C**). sHSPs are widely involved in various important cellular activities (**D**). The activity of sHSPs is regulated by phosphorylation cascades (**E**). Finally, abnormalities of sHSPs are closely associated with disease (**F**).

**Figure 2 cells-12-01947-f002:**
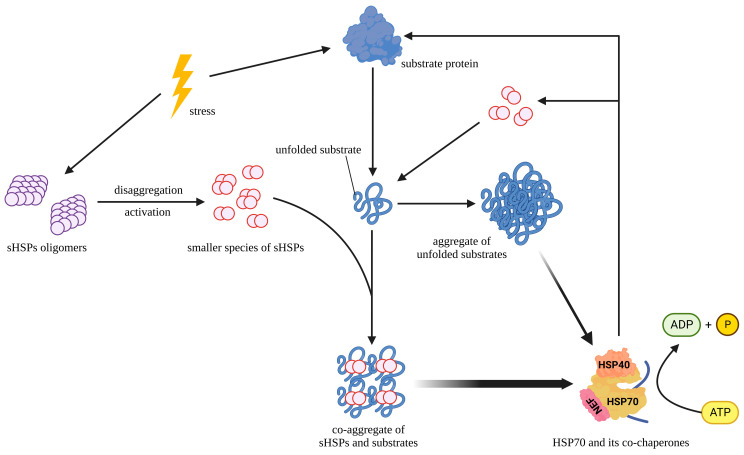
Role of sHSPs in preventing protein aggregation: The figure illustrates how sHSPs prevent the abnormal aggregation of proteins. Under stress conditions, substrate proteins unfold and aggregate. Meanwhile, sHSPs depolymerize from oligomers into smaller species, which subsequently co-aggregate with unfolded substrate proteins. The co-aggregates are smaller and more regular than those formed in the absence of sHSPs. In addition, the co-aggregates facilitate HSP70 and its co-chaperones to extract and refold substrate proteins via an ATP-dependent mechanism, as well as release and recycle sHSPs from the co-aggregates.

**Figure 3 cells-12-01947-f003:**
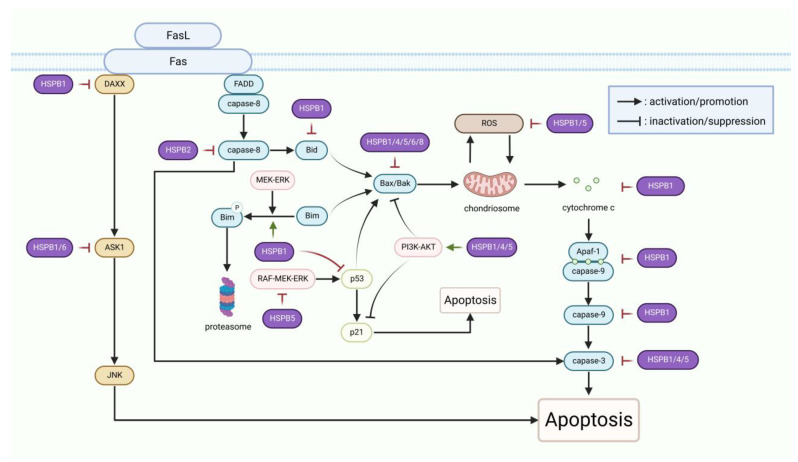
sHSPs participate in suppressing apoptosis: The figure illustrates the regulatory sites of sHSPs in the apoptotic signaling pathways. Purple represents sHSPs, blue represents proteins in caspase-dependent apoptotic signaling pathways, and “P” represents phosphorylation. The processes in the signaling pathways were simplified for sake of brevity.

**Figure 4 cells-12-01947-f004:**
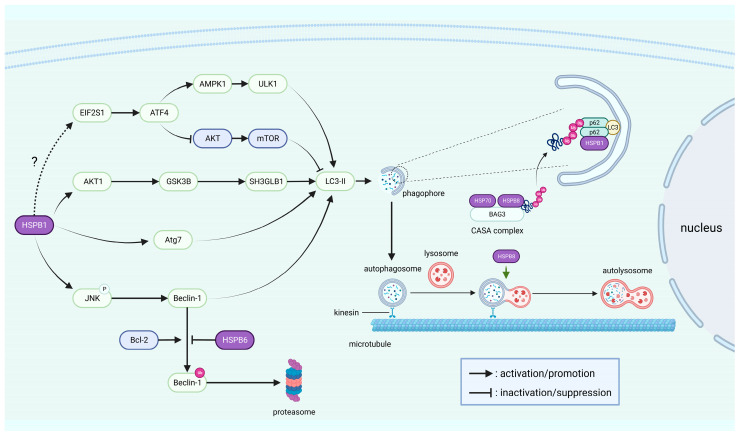
sHSPs are involved in the induction and progression of autophagy: The left half of the figure outlines the signaling pathways regulated by sHSPs during autophagy. The right half of the figure illustrates the role of sHSPs in the induction and progression of autophagy. The dotted line indicates a still controversial relationship. “P” stands for phosphorylation, and “Ub” for ubiquitination.

**Figure 5 cells-12-01947-f005:**
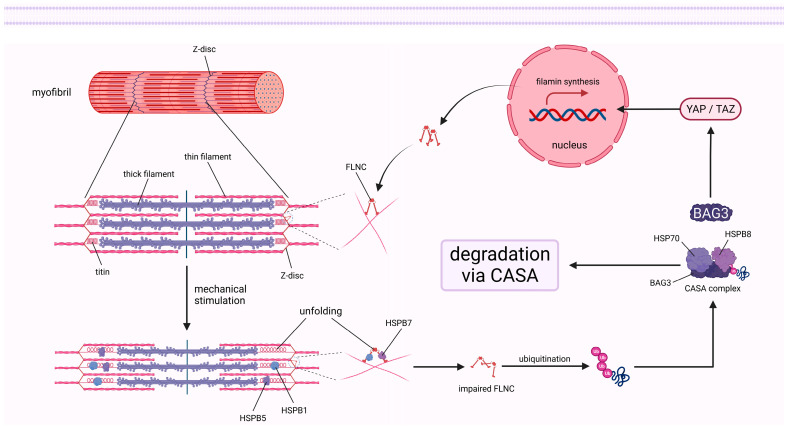
sHSPs respond to mechanical stimuli in myofibrils: The figure illustrates that sHSPs are involved in the protection of titin and FLNC, as well as the degradation and regeneration of damaged FLNC in myofibrils under mechanical stimulation.

**Figure 6 cells-12-01947-f006:**
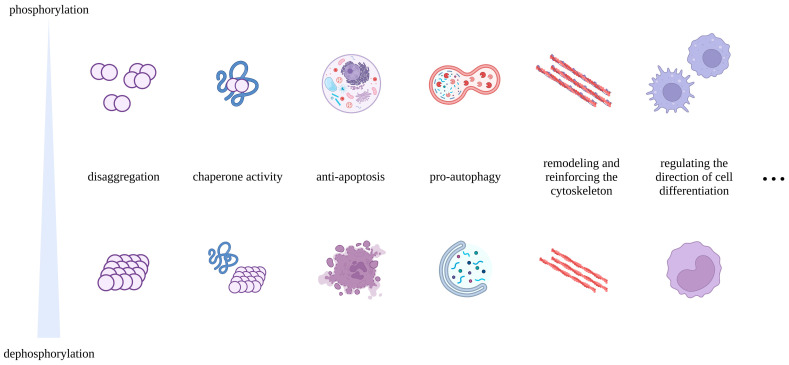
Effects of phosphorylation on the function of sHSPs: The figure outlines the contribution of phosphorylation to the function of sHSPs. The phosphorylated form is usually the active form of sHSPs, which induces oligomer depolymerization. The phosphorylation of sHSPs enhances their chaperone activity, promotes autophagy, favors the remodeling and strengthening of cytoskeleton, while preventing apoptosis and regulating the direction of cellular differentiation, as well as exerting other functions. Remarkably, the figure only shows a general trend in the effect of phosphorylation on the function of sHSPs, as the effects observed between different models are not always consistent.

**Table 1 cells-12-01947-t001:** Tissue distribution and function of mammalian HSPB1~10.

sHSPs	Alias	Tissue Distribution	Function	Main Ref.
HSPB1	HSP27 HSP25	Ubiquitous	Chaperone activity	[21]
			Anti-apoptosis	[22]
			Anti-ferroptosis	[23]
			Suppresses senescence	[24]
			Pro-autophagy	[25]
			Regulates cytoskeleton	[26]
			Maintains membrane stability	[27]
			Regulates differentiation	[28]
			Modulates angiogenesis	[29]
			Anti-inflammation	[30]
			Attenuates oxidative damage	[31]
HSPB2	MKBP	Cardiac and skeletal muscle	Chaperone activity	[32]
			Anti-apoptosis	[33]
HSPB3	HSPL27	Cardiac and skeletal muscle	Chaperone activity	[32]
			Promotes myogenesis	[34]
HSPB4	αA-crystallin	Eye lens	Chaperone activity	[32]
			Anti-apoptosis	[35]
			Anti-inflammation	[36]
HSPB5	αB-crystallin	Ubiquitous	Chaperone activity	[32]
			Anti-apoptosis	[37]
			Regulates cytoskeleton	[26]
			Maintains membrane stability	[38]
			Modulates angiogenesis	[39]
			Anti-inflammation	[39]
			Attenuates oxidative damage	[40]
HSPB6	HSP20	Ubiquitous	Chaperone activity	[32]
			Anti-apoptosis	[41]
			Pro-autophagy	[42]
			Regulates cytoskeleton	[43]
			Modulates angiogenesis	[44]
			Anti-inflammation	[41]
HSPB7	cvHSP	Cardiac and skeletal muscle, cardiovascular and insulin-sensitive tissues	Chaperone activity	[45]
			Regulates cytoskeleton	[46]
			Regulates differentiation	[47]
			Attenuates oxidative damage	[48]
HSPB8	HSP22	Ubiquitous	Chaperone activity	[32]
			Anti-apoptosis	[49]
			Pro-autophagy	[50]
			Regulates cytoskeleton	[51]
			Regulates differentiation	[52]
			Anti-inflammation	[53]
			Attenuates oxidative damage	[54]
HSPB9	CT51	Testis	Spermatogenesis	[55]
HSPB10	ODF1	Testis and kidney	Spermatogenesis	[55]
			Head-to-tail coupling of sperm	[56]

## Data Availability

Not applicable.

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
