# Peer review of "Functional Diversity of Mammalian Small Heat Shock Proteins: A Review"

_cells, 2023, doi:10.3390/cells12151947_

Round 1
Reviewer 1 Report
In this manuscript, the authors summarized the roles of sHSPs on regulation of cellular protein aggregation, apoptosis, autophagy, cytoskeleton assembly, cell differentiation, and oxidative damage. Phosphorylation may lead to sHSPs activation. Overall, the major body of this manuscript is informative for understanding sHSPs-mediated cellular activities.
Major points:
1) As oxidation, protein aggregation, apoptosis and autophagy are closely related, the authors may address the potential causes and effects and underlying mechanisms among these sHSPs-regulated cell activities.
Minor points:
2) Besides mutations, are there any studies on sHSPs expression levels and diseases occurrence?
The quality of English language is fine to me.
Reviewer 2 Report
This review provides an overview of the function of small heat shock proteins (sHSPs), with a focus on cellular activities. It also reviews some aspects of the role of sHSPs in different diseases. However, this reviewer feels that the literature review is not up-to-date and misses several significant studies. A substantial revision of this review would significantly increase the reader's interest. I have the following suggestions:
Table 1 has to be modified, and relevant references must be cited. This is essentially a copy of reference 19 from this MS.
Expand the role of: 1) sHSps in inflammation: there are several studies reported in the literature. 2) Mitochondrial function 3) Angiogenesis 4) Cancer 5) Senescence 6) Secreted sHSPs and mechanisms of secretion 7) Peptides derived from sHSPs and their function.
Include the therapeutic potential of sHSPs in different diseases. Several sHSPs have been the subject of in-depth research in a variety of disease models.
Perspectives must be adjusted to emphasize the areas that require further research.
N/A
Reviewer 3 Report
The paper of Guo et al. “Functional diversity of mammalian small heat shock proteins: a review” deals with important and interesting problem and contains valuable information concerning structure, mechanism of functioning and probable role of mammalian small heat shock protein in the cell. The Authors cited more than 180 papers containing information on the nature, distribution, oligomeric structure, posttranslational modification and participation of small heat shock proteins (sHsp) in different processes such as apoptosis, inflammation and oxidative stress, regulation of contractile apparatus and cytoskeleton, autophagy, cell differentiation and spermatogenesis. I am afraid that the main defect of this review is unsuccessful attempt to cover many diverse problems in comparatively small paper. Being restricted in paper size and at the same time trying to cover many diverse problems the Authors only briefly describe the data of literature without their explanation, critical analyses and detailed interpretation. Therefore, many important questions are discussed only superficially and incompletely correct.
The other points are as follows:
1. The central domain of small heat shock proteins (sHsp)is designated as alfa-crystallin domain, but not as crystal domain as it is indicated in summary and Introduction.
2. The sHsp undergo not only phosphorylation (as it is mentioned on lines 21-22 and 324-372) but many other posttranslational modifications such as glycosylation (DOI 10.1038/s41557-021-00648-8), acetylation (DOI 10.1074/jbc.M111.278549), deamidation (Asn/Asp conversion), Asp racemization (DOI 10.1002/pro.3821), SH-groups oxidation and modification (DOI 10.1074/jbc.M200591200) etc. All these modifications affect their structure and properties.
3. Lines 44-45 “sHsp have the ability to assemble into homo- or heteromeric oligomers to different degrees, usually with more than 12 subunits”. This is not completely correct since HspB6, HspB7, HspB8 are predominantly presented in dimeric or even monomeric forms.
4. Table 1. HspB7 is presented not only in cardiac and skeletal muscle, but also in adipocytes and other insulin-dependent tissues (DOI: 10.1074/jbc.274.51.36592, DOI: 10.1016/j.bbamcr. 2009.05.005.)
5. Lines 82-83. It is indicated “A recent study showed that HSPB1 interacts preferentially with 83 polyQ-expanded, mutant huntingtin protein (HTT) and affects its aggregation.” I suppose that HspB7, but not HspB1 interacts preferentially with huntigtin protein.
6. Lines 106-107 “Meanwhile sHsps depolymerize from oligomers into smaller species which consequently co-aggregate with unfolded substrate proteins”. I suppose that sHsp do not co-aggregate with substrate, but form soluble complexes that become good substrates for Hsp70 and other ATP-dependent chaperones.
7. The scheme, presented on Fig.3 is very complex. It is desirable to explain in more detail mechanism of sHsp action on each stage of apoptosis. Involvement of HspB4 expressed exclusively in eye lens in regulation of apoptosis should be more clearly explained.
8. The scheme on Fig.4 is again very complex. The Authors should clearly explain each of indicated pathways. The left part of this scheme deals with apoptosis, but not with autophagy as it is indicated in the title of this scheme. The Authors should clearly explain transition from Hsp70_HspB8_Bag3 complex to p62_HspB1_LC3 complex. What is the role of HspB1 located on microtubule?
9. Lines 242-246. It is stated “Previous studies revealed that unphosphorylated HspB1 forms large oligomers, which stoichiometrically bind to monomeric actin (G-actin) and inhibit actin polymerization 96,97. Subsequent research showed that phosphorylation of HspB1 by the p38/MK2 pathway induces the dissociation of oligomeric complexes, resulting in the release of HspB1-bound G-actin to promote actin assembly 98,99”. References 96 and 97 are reviews, but not experimental paper. Moreover, in review of Mounier and Arrigo (reference 97) it stated that interaction of HspB1 with actin is very complex and requires very complicated explanation. Indeed, in original paper of Benndorf et al. (JBC 269, (32), 20780-20784, 1994) it was shown that unphosphorylated monomers of Hsp25 were effective in inhibition of actin polymerization, whereas phosphorylated monomers or unphosphorylated oligomers were ineffective in regulation of this process. Moreover, intracellular concentration of sHsp is much smaller than that of actin and therefore formation of stoichiometric complex of G-actin and sHsp is highly improbable. At the same time there are experimental data indicating that Hsp27 is a weak F-actin side binding protein (DOI: 10.1155/2011/901572) that predominantly interacts with partially denatured, but not with native actin (DOI: 10.1111/j.1742-4658.2007.06117.x.)
10. Lines 260-261 “HSPB1 and HSPB7 have been shown to directly interact with FLNC, thus preventing the occurrence and progression of myopathy 106-108”. It remains unclear by what means interaction of two sHsp with filamin can prevent progression of myopathy.
11. Lines 267-268 “In addition, HSPB6 and HSPB7 modulate microfilaments by regulating adaptor proteins 14-3-3 and actin nucleator leiomodin 2, respectively 109,110”. It is incorrectly to compare leiomodin 2, a genuine actin-binding protein, with universal adaptor protein 14-3-3 that upon binding of phosphorylated HspB6 dissociates from still unknown protein that somehow regulates actin polymerization.
12. Description of Fig. 5 is unclear. What happens with filamin C after damage and what is the role of different sHsp in repairing of contractile apparatus? What is connection between Bag3 and sHsp and by what means Bag3 affects synthesis of filamin C in the cell? There is misprint of the word “filamin” in the right upper corner of Fig.5.
13. Lines 331-333 “Phosphorylation of HSPB1 is regulated by numerous kinases and pathways, among which the p38-MK2 and PI3K-AKT pathways as well as ERK1/2 have been widely reported in recent years 145”. In addition Hsp27 is phosphorylated by cGMP-dependent proteins kinase (see DOI: 10.1074/jbc.m009234200).
14. Lines 335-336 “Phosphorylation of HSPB8 at Ser24, Ser27, and Thr87 is regulated by PKC, casein kinase II, and ERK1 150,151.” Firstly, in all cases mentioned Hsp22 was phosphorylated under in vitro conditions. Secondly, it was shown that casein kinase can phosphorylate Hsp22 (reference 150) and it was postulated (but not proved) that this enzyme phosphorylates Ser47 and Thr176, but not the sites mentioned by Guo et al. in the text. Thirty, under in vitro condition cAMP-dependent protein kinase can also phosphorylate certain sites located in the N-terminal domain of Hsp22 (DOI: 10.1134/s0006297908020120).
15. Lines 347-349 “HSPB1 phosphorylation is imperative for its response to mechanical stress, which results in increased exposure of residues surrounding the phosphorylated site, facilitating the interaction of HSPB1 with client proteins 108” The data mentioned were obtained only on one substrate (fragment of filamin) and cannot be generalized to all client proteins.
16. On Fig.6 the Authors try to generalize effects of phosphorylation of different sHsp. I am afraid that this attempt is unsuccessful. sHsp form a very diverse group and phosphorylation can induce different effects on different members of this family. For instance, phosphorylation of HspB1 is accompanied by stabilization of cytoskeleton, whereas phosphorylation of HspB6 promotes smooth muscle relaxation and destabilization of cytoskeleton. Phosphorylation is not obligatory accompanied by increased chaperone-like activity etc.
Round 2
Reviewer 2 Report
The authors modified the MS satisfactorily.
Author Response
Dear Reviewer: We would like to express our great appreciation to you for your professional review work on our manuscript entitled “Functional Diversity of Mammalian Small Heat Shock Proteins: A Review” (Manuscript ID: cells-2500054) and your recognition of our revision. Your suggestions and comments have played an important role in enriching the content of our manuscript. In the second round of revisions, we have added some content about PTD-HSPB6 in section 4 “sHSPs and Disease” according to other reviewers’ suggestions (Lines 474-478). Revised portions are also highlighted in yellow in the revised version of our manuscript. In addition, we have modified some details of Figure 4, which can better depict that HSPB8 promotes the fusion of autophagosome and lysosome. Thank you again and best regards. Yours sincerely, Wei Yu Chaoguang Gu Correspondence: Wei Yu, Ph.D Phone: +86-0571-86843338 E-mail address: yuwei@zstu.edu.cn College of Life Sciences and Medicine, Zhejiang Sci-Tech University Xiasha High-Tech Zone No.2 Road, Hangzhou 310018, ChinaReviewer 3 Report
The Authors carefully considered all recommendations given by reviewer and provided detailed answer to practically all questions raised by reviewer. However, there are two points that require further improvement.
1. I asked to simplify complex schemes presented on Fig.3 and 4. The Authors have ignored this recommendation. Both these schemes contain many abbreviations that are not explained in text. If the Authors insist on including these complex schemes than they should provide detailed list of abbreviations at the beginning or at the end of their review.
2. The Authors extended the sixth chapter “sHsp and disease” and mentioned utilization of protein transduction domain for target delivering of HspB1. Actually, similar investigations were performed earlier with N-terminal domain of HspB6 (doi 10.3171/2009.7.JNS09730: doi: 10.1038/jid.2008.264)
Author Response
Dear Reviewer: We would like to express our great appreciation to you for your professional review work on our manuscript entitled “Functional Diversity of Mammalian Small Heat Shock Proteins: A Review” (Manuscript ID: cells-2500054) and your recognition of our revision. Your 16 constructive suggestions and comments have played an important role in improving our manuscript. In addition, we have made corrections according to new suggestions and comments, which we hope to meet with your approval. Revised portions are also highlighted in yellow in the revised version of our manuscript. The main corrections in the paper and the responses to reviewer’s comments are as following: 1. I asked to simplify complex schemes presented on Fig.3 and 4. The Authors have ignored this recommendation. Both these schemes contain many abbreviations that are not explained in text. If the Authors insist on including these complex schemes than they should provide detailed list of abbreviations at the beginning or at the end of their review. Response: We sincerely appreciate the valuable comments. However, in the original and revised version of our manuscript, we do have prepared a detailed list of abbreviations at the end of the article according to the format template. The list of abbreviations is located above the “References”, which may not be easy to be found. As your comments, we have checked the list of abbreviations again to make sure all the abbreviations were listed. (Lines 526-559) 2. The Authors extended the sixth chapter “sHsp and disease” and mentioned utilization of protein transduction domain for target delivering of HspB1. Actually, similar investigations were performed earlier with N-terminal domain of HspB6 (doi 10.3171/2009.7.JNS09730: doi: 10.1038/jid.2008.264) Response: We sincerely thank your careful reading. According to your comments, we have added relevant content and references in section 4 “sHSPs and Disease”. (Lines 474-478) In addition, we have modified some details of Figure 4, which can better depict that HSPB8 promotes the fusion of autophagosome and lysosome. Thank you again and best regards. Yours sincerely, Wei Yu Chaoguang Gu Correspondence: Wei Yu, Ph.D Phone: +86-0571-86843338 E-mail address: yuwei@zstu.edu.cn College of Life Sciences and Medicine, Zhejiang Sci-Tech University Xiasha High-Tech Zone No.2 Road, Hangzhou 310018, China